# The P2X7 Receptor and NLRP3 Axis in Non-Alcoholic Fatty Liver Disease: A Brief Review

**DOI:** 10.3390/cells9041047

**Published:** 2020-04-22

**Authors:** Marco Rossato, Angelo Di Vincenzo, Claudio Pagano, Hamza El Hadi, Roberto Vettor

**Affiliations:** 1Clinica Medica 3, Department of Medicine—DIMED, University of Padova, 35128 Padova, Italy; divincenzoang@gmail.com (A.D.V.); claudio.pagano@unipd.it (C.P.); dr.hamza.elhadi@gmail.com (H.E.H.); roberto.vettor@unipd.it (R.V.); 2Department of Medicine, Klinikum Rheine, 48431 Rheine, Germany

**Keywords:** ATP, P2X7 receptor, NAFLD, NASH, liver, fibrosis

## Abstract

Non-alcoholic fatty liver disease (NAFLD) is the most common liver disease worldwide, and its prevalence is reaching epidemic characteristics both in adults and in children. The increase of NAFLD prevalence parallels that of obesity, now representing the major cause of liver inflammation, increasing the risk of cirrhosis and hepatocarcinoma. Furthermore, NAFLD is a risk factor for cardiovascular diseases and type 2 diabetes, two of the major leading causes of morbidity and mortality in western countries. Thus a significant amount of studies have dealt with the evaluation of the possible molecular mechanisms leading to NAFLD and its inflammatory consequences within the liver, the non-alcoholic steatohepatitis, and cirrhosis. The inflammasome is a key player in the inflammation and fibrogenic responses in many different tissues, including the liver. The activation of the NLRP3 inflammasome requires the activation by extracellular adenosine tri-phosphate (ATP) of a specific purinergic receptor named P2X7 located in the target cells, although other pathways have been described. To this regard, extracellular ATP acts as an internal danger signal coming from damaged cells participating in the activation of the inflammatory process, a signaling pathway common to many different tissues. Here, we briefly review the involvement of the P2X7 receptor/inflammasome NLRP3 axis in the pathophysiological events leading to NAFLD and its inflammatory and fibrotic evolutions, reporting the possible therapeutical strategies targeting the P2X7 receptor/NLRP3 inflammasome.

## 1. Introduction

Non-alcoholic fatty liver disease (NAFLD) represents the most common liver disease and refers to a group of conditions characterized by excess fat accumulation within the liver of subjects who do not assume excessive alcoholic beverages and with the exclusion of other liver disorders such as viral or chemical hepatitis [1]. NAFLD affects about 25% of subjects worldwide, above all with industrialized economies, and is defined as the presence of at least 5% of liver steatosis without any sign of liver damage. Among subjects with NAFLD, usually showing no specific symptoms or signs, a small percentage will develop non-alcoholic steatohepatitis (NASH), a condition defined as the presence of at least 5% liver steatosis and inflammation with hepatocyte damage with raised liver enzymes in the blood and the possible presence of liver fibrosis [1]. As NAFLD stages progress, patients show an increased cardio-metabolic risk, diabetes mellitus, and metabolic syndrome, together with an increased risk of developing liver cirrhosis (2% of patients with NASH) [2] and also hepatocarcinoma (1–2% of patients with cirrhosis) [3], although it is not unusual that hepatocarcinoma develops in patients with non-cirrhotic NASH [4]. Even if NASH develops from an existing liver steatosis, it develops in a small fraction of patients with fatty liver [4]. In the pathogenesis of NASH, the role of lipids, lipid precursors and products of their metabolism are important as danger signals that may have a primary role in the starting and progression of the inflammatory process that ultimately leads to NASH, cirrhosis, and also to hepatocarcinoma. To this regard, the precise cascade of events regulating these processes has not been fully characterized as yet, although it is well accepted that hepatocyte injury (and death) is (are) conditions of primary importance to lead to NASH and liver fibrosis [5]. In fact, it is generally accepted that injured and dead cells release a number of factors that have been partially discovered, although not fully described so far, to be acting as danger signals leading to the activation of the immune cells and producing inflammation [6]. In general, inflammation can be activated by external micro-organisms stimulating adaptive and innate immunity. While the former is activated by a hypothetically indefinite number of antigens, the latter is activated by a small limited number of molecules collectively known as pathogen-associated molecular patterns (PAMPs) [7]. It is also well known that inflammation can also occur in the absence of external pathogens, defining the so-called “sterile inflammation” that can be initiated by internal danger signals normally located and hidden within the cell and released by damaged and dead cells in a substantially sterile milieu [8] and that are collectively known as danger-associated molecular patterns (DAMPs) [9].

## 2. Extracellular Adenosine Tri-Phosphate (ATP) as a DAMP

Many different DAMPs have been identified as endogenous molecules that can be released from damaged cells to activate sterile inflammation, from intracellular proteins (such as heat-shock proteins to non-proteic molecules that mostly do not share any common characteristic with each other [8]. To this regard, adenosine tri-phosphate (ATP), the energetic molecule normally stored at high concentrations within the cell cytoplasm, has been shown to be one of the most important DAMPs, given its peculiar properties: in fact, ATP is present at very low concentration in the extracellular space under basal conditions and is released at high concentrations after cell damage and death [10]. The concept of ATP acting as an extracellular signaling molecule comes from the pioneering hypothesis by Geoffrey Burnstock in the early 1970s [11], who first suggested that ATP can be sensed by the cells via specific plasma membrane receptors, which were discovered and cloned about twenty years later and that are grouped in two different subfamilies, the P2X and the P2Y purinergic receptors. The P2X purinergic receptors are ligand-gated cation channels comprising seven different subtypes that have been cloned and characterized and named P2X1-2-3-4-5-6-7 [12]. The P2Y purinergic receptors belong to the G-protein coupled receptors superfamily, comprising eight different subtypes named P2Y1-2-4-6-11-12-13-14 [12]. Along with its release after cell damage or death, ATP can also be actively released from many different living cell types via different mechanisms, including connexin/pannexin membrane hemichannels or specific membrane transporters [13].

The role of purines as signaling molecules has been clearly shown in the last decades, and now the role of the so-called purinergic signaling pathway in many different biological processes in invertebrate and mammalian cells is well established [12]. The purinergic signaling pathway encompasses different purine and pyrimidine nucleotides and the nucleoside adenosine and by different families of receptors comprising the P1 receptors, selective for adenosine and P2 receptors preferentially selective for ATP and ADP but sensitive also to other ATP-related nucleotides [14,15]. These receptors are expressed on cell surfaces and, substantially, all cells express the purinergic receptors, including the liver (Table 1).

The P1 purinergic receptors are G-protein coupled receptors binding adenosine and are grouped in four different subtypes named A1, A2a, A2b, and A3. All these receptors bind adenosine, with the A1 and A3 receptor subtypes coupled to the inhibition of adenylate cyclase via the activation of an inhibitory Gi protein, while the A2a and A2b receptor subtypes are coupled to the stimulation of adenylate cyclase via the activation of a stimulatory Gs protein.

The P2X family of purinergic receptors comprises seven subtypes of receptors that are activated each by a single physiological agonist and by different synthetic agonists (see Table 1) [16,17]. In humans, the P2X receptors are ligand-gated ion channels formed by subunits ranging from 379 to 595 amino acids with two transmembrane domains and one extracellular ectodomain binding ATP. The P2X receptors are formed by homotrimers or heterotrimers being functionally active when three subunits (homo- or hetero-trimers) are bound together [18]. The activation of the P2X receptors needs the binding of at least three ATP molecules, also considering that the different P2X receptors show a different affinity for ATP [18]. The activation of the P2X receptors leads to the opening of a non-selective ion channel permeable to small cations (Na^+^, K^+^, Ca^2+^) or, depending on ATP concentration and on the P2X receptor subtype, to the opening of a plasma membrane pore permeable to small molecules with an MW up to 900 Da.

The P2Y purinergic receptors belong to the G-protein coupled receptors superfamily with the classical structure of seven-transmembrane domains with an amino acid length ranging from 328 to 377 [18]. Their activation leads to Gi/Gq-mediated signaling with the involvement of different intracellular messengers [19]. At variance with the P2X receptors, the P2Y receptors are activated not only by ATP but also, and in some cases, preferentially by other purines and pyrimidines (ADP, UTP, and UDP; see Table 1).

Purinergic receptors have evolved in all organisms from protozoa to man, thus underlining the importance of such signaling systems in the physiology of living organisms [12]. In particular, although all cells express P2 purinergic receptors, it is well known that P2 purinergic receptors are mainly expressed in the cells of the immune system participating in the complex mechanisms regulating the inflammatory process [14].

There is no doubt that the signaling systems controlling the response of living organisms to external pathogens are of primary importance, and to this regard, the regulation of the immune system and inflammatory response are vital for higher mammalians. Among the different purinergic receptors subtypes, the P2X7 receptor is undoubtedly mainly involved in the regulation of the inflammatory process [18].

Here we briefly describe the role of extracellular ATP and P2X7 receptors in the activation and modulation of the inflammatory process that characterizes fatty liver disease, NASH, and liver fibrosis.

## 3. ATP as an Extracellular Mediator of Inflammation

The cytoplasm of most mammalian cells contains ATP in concentrations as high as 5–10 mM, while its concentrations in the extracellular space are around the high nanomolar/low micromolar range [12]. Once released from the cytoplasm of damaged/dead cells or during its active release, ATP can reach high concentrations in the extracellular space, thus activating its receptors in the target cells. Furthermore, in the extracellular space, ATP is metabolized by ectonucleotidases to ADP, AMP, and adenosine molecules functioning as ligands of other purinergic receptors, as described above. The P2X7 receptor has been clearly involved in inflammation and is the most studied purinergic receptor subtype regulating the inflammatory process [20]. The P2X7 receptor is an extracellular ATP-gated ion channel permeable to small cations (Na^+^, K^+^, and Ca^2+^), although, at high ATP concentrations (> 0.3–0.5 mM), it determines the opening of a non-selective pore that is permeable to small solutes of mw up to 900 Da [21]. The P2X7 receptor is expressed by all immune and inflammatory cells, being upregulated during inflammation [12,20]. As reported before, ATP can be considered the prototype of what DAMPs are since it is released from a damaged cell, and, once in the extracellular space, it can start and amplify the inflammatory process by activating many different purinergic receptors in the target cells. Among these receptors, the P2X7 is the main involved in the pro-inflammatory effects elicited by the interaction with ATP. These potent pro-inflammatory effects are due to the peculiar ability of the P2X7 receptor, once activated by ATP in the induction of processing, maturation, and secretion of IL-1β, probably the most potent pro-inflammatory cytokine [21].

## 4. Ectonucleotidases

Under basal conditions, the ATP concentration in the extracellular space is almost undetectable. The role of ATP as an ideal extracellular signaling molecule is due to its peculiar ability to be undetectable under resting conditions being stored within the cells and released under certain circumstances, thus activating the P2 receptors on target cells and then returning to basal concentrations in the extracellular space after its release. This is due to the activity of a series of enzymes named ecto-nucleotidases, including ecto-nucleoside triphosphate diphosphohydrolases (ENTPDases), ecto-5-nucleotidases (NT5E)/CD73, ecto-nucleotide pyrophosphate phospho- diesterases (E-NPPs), CD38/NADase, nucleoside diphosphate kinase, and others [22]. These enzymes are involved in purine and pyrimidine metabolism, converting ATP in ADP and AMP and AMP to adenosine. Thus ATP exerts its functions through the activation of ATP sensitive P2 purinergic receptors, but the products of its metabolism also activate other P2 and P1 purinergic receptors. Ectonucleotidases are constitutively expressed in many different cells, and it is known that some of them, namely, CD39 and CD73, can be overexpressed in cells of the immune system after exposure to inflammatory cytokines, oxidative stress, and hypoxia [22]. Previous studies have reported the possible involvement of different ectonucleotidases in liver steatosis and fibrosis, although with different results. In fact, while the deletion of CD39 induced hepatic insulin resistance and altered glucose metabolism and liver steatosis [23], the deletion of CD73 is protective with respect to induced liver fibrosis [24]. Furthermore, active Entpd2 protects versus chemical-induced liver fibrosis; in another study, Entpd2 deletion was without effects in liver-induced fibrosis [25]. Nonetheless, there is evidence that these enzymes are involved in the modulation of the effects of the different purines and pyrimidines on target cells and within the liver in particular, possibly participating to the different steps leading to liver steatosis, NASH, fibrosis, and also to hepatocarcinoma [22,26].

## 5. The NLRP3 Inflammasome

Following liver injury, different inflammatory cytokines are expressed and released in the liver, causing inflammation and fibrosis. Among these inflammatory cytokines, IL-1β is one of the most important. The production and secretion of IL-1β are under the control of the so-called inflammasome, a large cytoplasmic multimeric complex sensing PAMPs/DAMPs via the activation of a sensory molecule that is a member of the family of nucleotide binding oligomerization domain (NOD)-like receptor (NLRs) [27]. Generally, this multiprotein complex is composed of an NLR, an adapter protein named apoptosis-associated speck-like protein containing CARD (ASC), and the enzyme caspase-1 [28]. Based on their molecular structure and function, NOD-like receptors can be grouped into three different subfamilies: NLRP, NOD, and ICE-protease-activating factor/neuronal apoptosis inhibitory protein [29,30]. NLRs participate in the formation of inflammasomes that have been named based on NLRs classification [29]. The NLR couples with the pro-caspase 1, leading to its activation to caspase 1, thus activating the enzymatic maturation of important pro-inflammatory cytokines such as pro-IL-1β and its mature form IL-1β [31]. Pro-caspase 1 is constitutively expressed within the cell being inactive until its activation is requested. Activation of caspase 1, together with IL-1β, is also coupled to the production of IL-18, IL-1alpha, and fibroblast growth factor (FGF)-21 [32]. Many different inflammasomes have been described that are activated by different danger signals, including NLRP1, NLRP2, NLRP3, NLRP6, NLRC4, AIM2, and RIGI [28]. Among the different NLRs, the cytosolic sensor NLR family pyrin domain containing 3 (NLRP3) inflammasome is one of the most studied and has been shown to respond to PAMPs and DAMPs, and, in particular, to extracellular ATP [33]. On the contrary, it is known that NLRP3 inflammasome activity is inhibited by type-1 interferon and nitric oxide [28]. The liver expresses many different types of inflammasomes (namely NLRP1, NLRP2, NLRP3, NLRP6, NLRC4, and others) [28]. Furthermore, within the liver hepatocytes, stellate cells, endothelial cells, Kupffer cells, and liver fibroblasts express inflammasome components, caspase-1 and ASC, and also the P2X7 receptor [28]. Thus it seems that the activation of the inflammasome in the liver plays a primary role in the inflammatory process in NASH. These inflammasomes play a primary role in inflammation since, in experimental conditions knocking down inflammasomes, the inflammation process is blunted [8].

## 6. P2X7 Receptor, NLRP3 Inflammasome, NAFLD and Liver Fibrosis

NAFLD is characterized at the beginning by the deposition of lipids within the liver in subjects not assuming alcohol and without any other known cause of liver damage. The progression from this “simple” liver steatosis to NASH and fibrosis involves many different mechanisms that have not been completely discovered yet. However, as detailed above, it seems that the inflammasome has a primary role in this progression in the liver, where the different components of the NLRP3 inflammasome are expressed and increased during liver damage [34]. Thus it is now well accepted that this multiprotein complex plays a pathogenic role in NASH and liver fibrosis, and different studies support this hypothesis. In particular, studies with NLRP3 knock-out mice showed that these animals have lower inflammation and fibrosis, while studies with NLRP3 knock-in mice showed higher inflammation and fibrosis [28]. Other studies evaluating the role of caspase-1 and IL-1β knock-out animals have demonstrated the animals to be protected from NASH and liver fibrosis induced by high-fat diets [28,35,36,37]. In agreement with this observation, in vitro studies have shown that IL-1β treatment induced lipid accumulation in hepatocytes and the activation of fibrogenesis in hepatic stellate cells [28,37]. In the liver, hepatocytes, stellate, and Kupffer cells all express the NLRP3 inflammasome being activated after liver injury to determine the maturation and release of IL-1β, IL-18, and the following cascade of other inflammatory cytokines, favoring liver inflammation and fibrosis [38].

Previous studies have shown that extracellular ATP is a strong inducer of the activation of NLRP3 activation and of maturation and release of mature IL-1β in mouse macrophages [12]. Initially described as expressed exclusively in the cells of the immune system [39], the P2X7 receptor is now known as expressed almost ubiquitously, although cells of the immune system express this receptor to the highest levels [12]. The P2X7 receptor is a ligand-gated ion channel permeable to small cations such as K^+^, and it is the K^+^ efflux from the cytoplasm that represents the key signaling pathway leading to caspase-1 activation, maturation of pro-IL-1β, and mature IL-1β release in macrophages [40]. Kupffer cells are considered the liver resident macrophages and are particularly abundant within the liver. It has been previously demonstrated that Kupffer cells express the P2X7 receptor, which activation seems to have a role during the release of ATP from damaged cells in different liver damage models [41]. Furthermore, the liver internal macrophages represented by the Kupffer cells can be amplified by recruiting circulating monocytes as a consequence of liver damage and the release by hepatocytes, stellate cells, and Kupffer cells themselves of specific cytokines such as CCL2 acting as a chemoattractant [42]. This process is able to rapidly expand the macrophage population to protect the liver but possibly leading also to excessive inflammatory/reparative responses [42].

The discovery of the NLRP3 inflammasome, together with the role of ATP and P2X7 receptors in the processing and release of mature IL-1β, has provided the connection between extracellular ATP, P2X7 receptor, NLRP3 inflammasome, and IL-1β release and the activation and amplification of the inflammatory process. To this regard NLRP3 activation has been described as a “two-step” process: the first is via an NFkB activator as well as LPS, while the second step is mainly via the so-called DAMPs, distinct molecules released by damaged cells acting as alert signals to trigger the innate immune response [38]. As detailed above, among the different DAMPs extracellular ATP (via the activation of the P2X7 receptor on target cells) is one of the most studied, although other activators have been reported, such as reactive oxygen species (ROS), crystals, or large particles [27,34].

Inflammasome has been shown to be involved in a huge number of biological processes such as fibrosis [43], as demonstrated by experiments showing that mice lacking inflammasome components have reduced fibrosis (see [31] and the references therein). Tissue fibrosis is characterized by an increase of fibroblast proliferation, increased production of extracellular matrix proteins leading to the substitution of normal tissue with fibrotic tissue [31]. To this regard, IL-1β has been shown to stimulate transforming-growth factor (TGF)-β and collagen production leading to tissue fibrosis [31,43]. Considering the role of the P2X7R and inflammasome in the triggering of the inflammatory process and expression of activators of fibrosis, it is possible that this pathway may have a primary role in the inflammatory process and of liver fibrosis/cirrhosis in NASH.

The evolution of NASH to fibrosis regards about 2% of subjects with NASH and is due to the production and accumulation of extracellular matrix protein induced by the activation of hepatic stellate cells by cytokines with fibrogenic activity such as TGF-1β [44,45]. Together with TGF-1β, IL-1β has also been shown to participate in liver fibrosis [46]. In different experimental animal models of liver fibrosis (induced by toxic molecules such as carbon tetrachloride or by common bile duct ligation), the blockade of the P2X7 receptor significantly reduced the expression of the pro-fibrotic cytokines and liver fibrosis [47,48]. Furthermore, in two animal models of NASH (induced by a high-fat diet or by a methionine choline-deficient diet), the deletion of the P2X7 receptor gene or the use of P2X7 receptor antagonist, respectively, showed a reduction of liver toxicity [49,50]. Furthermore, many different observations have suggested that NLRP3 is also involved in NASH-liver fibrosis evolution: the deletion of the NLRP3 gene or the pharmacological blockade of NLRP3 inflammasome reduced liver fibrosis in animal models of NAFLD/NASH [51,52]. Finally, also liver fat accumulation in mice fed with a high-fat diet was lower when the P2X7 receptor/NLRP3 axis was blocked, to underline the role of this pathway from the early stages of NAFLD [53].

To this regard, within the liver, many different cell types express NLRP3 inflammasome, from hepatocytes to Kupffer and stellate cells and endothelial cells [27,28], further confirming the reported observations. In particular, stellate cells represent the mesenchymal liver cells located in the Disse space, playing a primary role in the fibrogenetic process within the liver, promoting the excessive accumulation of extracellular matrix and type I collagen leading to liver fibrosis [38,54]. To this regard, it has been previously shown that P2X7R is involved in ATP activation of stellate cells via activation of NLRP3 inflammasome [55] and might also mediate the activation of stellate cells by leptin in non-alcoholic steatohepatitis [56].

## 7. P2X7 Receptor as a Therapeutic Target in NASH and Liver Fibrosis

Given the important role of the purinergic signaling pathway in the inflammatory process in liver steatosis, NASH and liver fibrosis, the possibility to perturbate this pathway could lead to the discovery of new therapeutic strategies in these liver diseases. This purinergic signaling pathway encompasses ATP, P1, and P2 receptors, ectonucleotidases all participating in the activation, mainteinance, amplification, and extinguishing of the inflammatory signal activated by extracellular ATP. Thus, as depicted in Figure 1, the damaging of hepatocytes by lipids and lipid metabolism products accumulation could lead to the release of ATP in the extracellular space; activation of the P2 purinergic receptors on near cells (namely Kupffer and stellate cells) leading to activation of inflammation in particular via the activation of the P2X7 receptor and NLRP3 inflammasome [14,55,56]. Thus the targeting of this purinergic pathway might be a promising therapeutical strategy to extinguish the inflammatory process activated after liver injury. Since the blockade of the P2X7 receptor in experimental animals (via P2X7 receptor gene deletion or using pharmacological agents) results in reduced fat accumulation in the liver, reduced NLRP3 activation and IL-1β production and release, reduced inflammation, reduced liver steatosis evolution to NASH, and reduced liver fibrosis, it is possible to hypothesize that the P2X7 receptor/NLRP3 axis might be an effective therapeutical target to tackle liver steatosis and the fibrotic evolution of NASH [38]. A number of different pharmacological molecules and biological agents have been developed to block the P2X7 receptor/NLRP3 axis [57,58,59] and are actively utilized in experimental studies in many different diseases in humans, from rheumatoid arthritis to chronic obstructive pulmonary disease and Crohn’s disease (see [12] and the references therein). To date, the clinical efficacy in the reported studies has not been so favorable, although the blockade of some downstream effectors such as IL-1β induces important clinical results [12]. In the evaluation of the reasons of a poor clinical response using the blockade of the P2X7 receptor, many different aspects have to be taken into consideration, starting from the possible ineffectiveness of P2X7 receptor inhibition for each specific disease in humans [12], up to differences in the genetic background affecting the response to the P2X7 receptor block [60], together with the specific physico-chemical interaction with the P2X7 receptor of each receptor antagonist in the different activating conditions [61]. Further studies will be necessary to strongly confirm in humans the observations coming from animal studies and to tailor the proper P2X7 receptor antagonist for each specific pathological condition and probably for specific groups of patients.

## 8. Conclusions

The purinergic signaling pathway is a highly complex system playing a major role in inflammatory processes, modulating the activation in many different tissues where it regulates the maintenance and the finishing of inflammation. The role of ATP released by damaged cells in the extracellular space, the activation of the P2X7 receptor, together with the involvement of NLRP3 inflammasome and pro-inflammatory cytokines production is now also well-proven in the liver, with substantial evidence showing the involvement of the P2X7 receptor/NLRP3 inflammasome in the induction of NAFLD/NASH and liver fibrosis, at least in animal models, involving the modulation of hepatocytes, and Kupffer and stellate cell activity. In particular, in liver steatosis, lipid accumulation could induce hepatocytes damage with ATP release that, in turn, can activate the P2X7 receptor on Kupffer and stellate cells with the secretion of many different pro-inflammatory and pro-fibrotic cytokines. The demonstration that this pro-inflammatory axis is also involved in human NALD/NASH evolution and the detailed understanding of the activation of this ATP/P2X7 receptor/NLRP3 inflammasome signaling pathway could offer the possibility to disclosure the NASH pathophysiology, thus possibly leading to the pharmacological targeting of the specific components of this signaling pathway, mainly the P2X7 receptor or its downstream effectors to modulate the pathophysiological events characterizing the NAFLD spectrum of disease that is reaching epidemic characteristics worldwide. Furthermore, the ectonucleotidases favor the metabolism of the ATP released by damaged cells, generating ADP and AMP, thus progressively reducing the ATP concentration outside the cells on one side and activating many other different purinergic receptors sensitive to ADP and AMP favoring and/or counteracting the effects of ATP or liver inflammation. These different purinergic receptors, of which the P2X7 is the most studied, together with NALP3 inflammasome and the downstream signaling pathway, could be potential targets for different molecules that could represent a possible future therapy to modulate different conditions in the liver.

As briefly reported in the present review, the liver and its different cell populations express different purinergic receptors playing different roles in activating and positively or negatively modulating the inflammatory response and the pro-fibrotic processes. 

Further studies will be necessary to understand the precise players of the purinergic system within the liver in steatosis, NASH, and liver fibrosis to discover specific treatments aimed to perturb this important signaling pathway.

## Figures and Tables

**Figure 1 cells-09-01047-f001:**
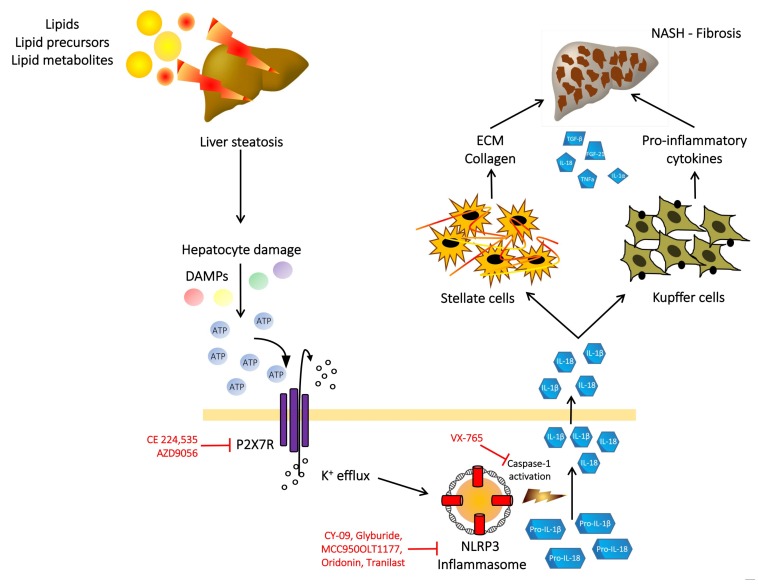
Schematic illustration of the P2X7 receptor/NLRP3 inflammasome axis involvement in non-alcoholic fatty liver disease (NAFLD) pathogenesis and evolution to non-alcoholic steatohepatitis (NASH) and liver fibrosis. Upon exposure to an injury (for example due to lipid accumulation), damaged or dead hepatocytes release molecules (such as adenosine tri-phosphate - ATP) acting as danger-associated molecular patterns (DAMPs) on the P2X7R receptor which activation determines the assembly and activation of the NLRP3 inflammasome and the following cascade leading to caspase-1 activation, pro-IL 1β and pro-IL 18 cleavage and mature IL-1β and Il-18 release. These inflammatory cytokines modulate the activity of the stellate and Kupffer cells, further amplifying the inflammatory process, determining NASH and liver fibrosis due to extracellular matrix and collagen deposition. In red are reported the possible therapeutical targets to inhibit the inflammatory response due to P2X7 receptor/NLRP3 inflammasome axis activation.

**Table 1 cells-09-01047-t001:** Classification of purinergic receptors.

Purinergic Receptor		Endogenous Agonist	Expression in Normal Liver
P1 receptors	A1	Adenosine	Yes
	A2a	Adenosine	Yes
	A2b	Adenosine	Yes
	A3	Adenosine	Yes
P2X receptors	P2X1	ATP	Yes
	P2X2	ATP	Yes
	P2X3	ATP	Yes
	P2X4	ATP	Yes
	P2X5	ATP	No
	P2X6	ATP	Yes
	P2X7	ATP	Yes
P2Y receptors	P2Y1	ADP	Low
	P2Y2	ATP, UTP	Yes
	P2Y4	ATP, UTP	Yes
	P2Y6	UDP	Yes
	P2Y11	ATP	Yes
	P2Y12	ADP	Low
	P2Y13	ADP	Low
	P2Y14	UDP	Low

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
