# Peer review of "The P2X7 Receptor and NLRP3 Axis in Non-Alcoholic Fatty Liver Disease: A Brief Review"

_cells, 2020, doi:10.3390/cells9041047_

Round 1
Reviewer 1 Report
In this review article, Rossato et al., have provided information on NAFLD, P2X/P2Y receptors, NLRP3 inflammasome and liver inflammation. This review although different from previously published studies do not really provide any new information or in other words do not critically review available literature in the field. Overall, this is do not look like a review. I think the whole review needs to be restructured and re-written.
- Abstract looks fine but does not reflect the information provided in the review.
- At least one figure showing the inflammasome, P2X7 receptor, ATP, inflammation in liver cells would help the readers understanding the point of this review. Including therapeutics in the figure itself would be helpful self-explanatory.
- Authors should give a perspective or analysis of what is known, what is being done, and how the conclusions from previous studies are justified.
- The concluding remarks should highlight some of the crucial unanswered questions in the field, so as to direct the readers to the main areas to be focused in the future research in the field.
- Many important references are missing. Whole reference section is confusing probably because [1] is given to reference. The authors need to be extremely careful with this. Cited reference [27] and context is completely wrong.
Author Response
We thank the Reviewer for his/her suggestions that have contributed to improve the revised version above all with the suggestion to insert a figure.
1.The abstract has been slightly reviewed.
2.We have added a figure reporting the mechanisms regulating the activation of the P2X7 receptor/NLRP3 inflammasome within the damaged liver together to the docking points for putative therapeutical approaches.
3 and 4. We have temptatively tried to follow the Reviewer’s suggestion above all in the concluding remarks.
5. The Reviewer is correct. This mistake has completely disrupted the order of the numbered references. This has been amended. The reference [27] cited by the Reviewer was indeed the [26] due to the fact that the reference [1] was addressed to References and this for all the cited references reported an altered (plus one) number. New references have been added (7,55-59).
Reviewer 2 Report
The review by Rossato et al. (Cells – 729538) has been designed in order to offer a brief overview on the available data on the involvement of the P2X7 receptor/inflammasome NLRP3 axis in the progression of NAFLD. In this review Authors first offer an introduction emphasizing the very significant impact of this disease in the general population particularly of developed countries. Authors then offer an overview on ATP, P2X7 receptor and NLRP3 axis to then dedicate the last two sections to the putative role of this axis in NAFLD and liver fibrosis (section 6) and to P2X7 receptor as potential target for therapeutic strategies designed to interfere with NAFLD progression (i.e., pathogenesis of steatosis and inflammatory and fibrogenic progression to NASH and more advanced conditions of liver disease).
The proposed review is a honest, appreciable and up-to date attempt to overview the specific involvement of the P2X7 receptor/inflammasome NLRP3 axis in the progression of NAFLD. The matter is of potential interest although it should be noted that research and clinical data available are at present quite limited on this topic. I can offer the following few comments.
- I would first suggest to modify the actual title of the review since the manuscript is really all centered on NAFLD.
- Data on the specific topic of this review are indeed limited but Author may include data from some references (and the messages therein) that are not mentioned in the actual version of the manuscript. Some examples (not exaustuve) are the following:
- Chandrashekaran V, Das S, Seth RK, Dattaroy D, Alhasson F, Michelotti G, Nagarkatti M, Nagarkatti P, Diehl AM, Chatterjee S. Purinergic receptor X7 mediates leptin induced GLUT4 function in stellate cells in nonalcoholic steatohepatitis. Biochim Biophys Acta. 2016 Jan;1862(1):32-45. doi: 10.1016/j.bbadis.2015.10.009.
- Jiang S, Zhang Y, Zheng JH, Li X, Yao YL, Wu YL, Song SZ, Sun P, Nan JX, Lian LH. Potentiation of hepatic stellate cell activation by extracellular ATP is dependent on P2X7R-mediated NLRP3 inflammasome activation. Pharmacol Res. 2017 Mar;117:82-93. doi: 10.1016/j.phrs.2016.11.040.
- Mehal WZ. The inflammasome in liver injury and non-alcoholic fatty liver disease. Dig Dis. 2014;32(5):507-15. doi: 10.1159/000360495.
- I would suggest Authors to expand data and concepts related to the role of NLRP3 in NAFLD progression; this is an area that should deserve more attention (there are more data available in the literature to be cited).
Author Response
We thank the Reviewer for her/his appreciation of our paper.
1.Following the suggestions we have modified the title as requested.
2.We have reviewed the suggested references reporting few comments in the text (as detailed in the revised version). Those references have been added to the specific section.
3.Considering the last suggestion of the reviewer, we honestly think that the subchapter 6 appears exhaustive as it stands without being repetitive and redundant to give a description of the role of NLRP3 in NAFLD progression to NASH and fibrosis.
Round 2
Reviewer 1 Report
The review has been substantially improved and figure addition has definitely helped communicating the ideas of the authors. I only have minor comments:
- The manuscript should be proof read for errors.
- DAMPs stands for "Damage Associated Molecular Patterns" and not "Danger associated...."